# Should Cystatin C eGFR Become Routine Clinical Practice?

**DOI:** 10.3390/biom13071075

**Published:** 2023-07-05

**Authors:** Sebastian Spencer, Robert Desborough, Sunil Bhandari

**Affiliations:** 1School of Medical Sciences, University of Hull, Hull HU6 7RX, UK; 2Hull York Medical School, University of Hull, Hull HU6 7RU, UKsunil.bhandari@nhs.net (S.B.); 3Academic Renal Research, Hull University Teaching Hospitals NHS Trust, Hull HU3 2JZ, UK

**Keywords:** chronic kidney disease, cardiovascular disease, cystatin C, GFR

## Abstract

Kidney function assessment is crucial for diagnosing and managing kidney diseases. Glomerular filtration rate (GFR) is widely used as an indicator of kidney function, but its direct measurement is challenging. Serum creatinine, a commonly used marker for estimating GFR (eGFR), has limitations in accuracy and sensitivity. Cystatin C, a protein freely filtered by the glomerulus, has emerged as a promising alternative marker for kidney function. It is unaffected by muscle mass and shows stronger associations with cardiovascular disease and mortality than creatinine. Various equations have been developed to estimate GFR using creatinine or cystatin C alone or in combination. The CKD-EPI_creat-cys_ equation combining both markers demonstrates improved accuracy in GFR estimation, especially for individuals with eGFR values of 45–59 mL/min/1.73 m^2^. Cystatin C-based estimates of GFR outperform creatinine-based estimates in predicting clinical outcomes and identifying patients at higher risk, particularly in elderly and non-white ethnic groups. Cystatin C offers advantages over creatinine as a marker of kidney function. It is not influenced by non-kidney factors and provides more accurate estimation of GFR, aiding in the early detection of kidney disease and predicting adverse outcomes. Incorporating cystatin C into routine kidney function assessment may improve patient risk stratification and guide clinical decision-making. However, widespread adoption of cystatin C testing requires increased availability and accessibility in clinical laboratories. Further research and implementation efforts are needed to fully realize the potential of cystatin C in kidney function assessment and improving patient outcomes.

## 1. Introduction

Kidney function can, especially amongst nephrologists, refer to a whole variety of biological functions of the kidney: excretion, filtration or clearance as well as metabolic or endocrine physiological roles. However, following the discovery that “pure” filtration occurs at the glomerulus of the nephron, the glomerular filtration rate (GFR) is often considered the best overall index of kidney function in health and disease. Unfortunately, true GFR cannot be measured directly and requires the assessment of the clearance of either exogenous or endogenous filtration markers. These markers should have particular chemical and physiological qualities to allow for an accurate and reliably measured GFR. That is, to be as close to the true GFR as possible. Whether they be exogenous or endogenous, all should be chemically inert in vivo, freely filtered by the glomerulus, neither secreted nor reabsorbed by the tubules and eliminated solely by the kidneys [1].

Whilst there are many suggested markers for estimating the glomerular filtration rate (eGFR)—such as beta-trace protein, beta2-microglobulin and cystatin C, which has garnered lots of interest in recent years—the most common method for calculating eGFR worldwide is through creatinine measurement (eGFRcr), which has become more reliable with standardization to isotope dilution mass spectrometry (IDMS)-traceable techniques and the development of more robust eGFR equations. However, it is an imperfect tool for estimating GFR as it tends to overestimate GFR and underestimate the presence or severity of chronic kidney disease (CKD) in older individuals or those with lower-than-average muscle mass for their age. Moreover, it is insensitive in detecting early kidney disease. Serum creatinine measurement and the subsequent calculation of eGFR through derived equations have long been the mainstay of laboratory assessments of GFR; however, creatinine is not an ideal exogenous marker of GFR, being only freely filtered. As new creatinine-based equations have been developed and become more accurate, parallel efforts have been in play to identify, verify and validate new exogenous markers and the eGFR equations derived.

The Kidney Disease Improving Global Outcomes (KDIGO) guidelines endorse the use of cystatin C or a clearance measurement for confirmatory testing in specific circumstances when the estimated glomerular filtration rate based on serum creatinine is less accurate [2] and when precision is required for dosing; otherwise, estimates may be unreliable due to the extremes of muscle mass [3]. For adults with eGFR_creat_ 45–59 mL/min/1.73 m^2^ who do not have markers of kidney damage, measuring cystatin C is suggested to confirm the diagnosis of CKD. If the eGFR_cys_ or combination eGFR_creat-cys_ (creatinine and cystatin) is <60 mL/min/1.73 m^2^, the diagnosis of CKD is confirmed, but if it is >60 mL/min/1.73 m^2^, the diagnosis of CKD is not confirmed. However, the National Institute for Health and Care Excellence (NICE) acknowledges the lack of high-quality evidence and currently does not make positive recommendations for the use of cystatin C equations to estimate GFR in routine clinical practice due to the discord between eGFR_creat_ and eGFR_cys_, particularly in older adults [4]. However, eGFR_cys_ has been shown to have a stronger correlation with future risk of cardiovascular disease and death than eGFR_creat_ [5,6]. Despite this, it is uncommon in clinical practice to conduct concordance testing for CKD using cystatin C, as recommended [7].

Chronic kidney disease affects a considerable proportion of the population, with an estimated 3.5 million adults in the UK alone being diagnosed [4]. Population-based studies have shown that there is an increased risk of death and cardiovascular mortality as the eGFR falls below 60 mL/min/1.73 m^2^, or when albumin is detected in urinalysis [8]. People diagnosed with CKD are more likely to experience cardiovascular disease than to progress to end-stage kidney failure (ESKF or GFR < 15 mL/min/1.73 m^2^) [9]. The management of modifiable cardiovascular risk factors, such as improved blood pressure and tighter glycaemic control in people with diabetes mellitus, can reduce the progression of CKD to more advanced stages [10]. There are specific risk factors associated with advanced CKD that contribute to the high rates of morbidity and mortality even at younger ages [11], for example, there is a correlation as proteinuria levels rise, the risk of mortality and myocardial infarction rises regardless of the eGFR level [12].

The increased risk seen in people with CKD cannot be explained by traditional risk factors alone as numerous studies have shown that even low levels of urinary albumin are associated with a higher risk of cardiovascular disease (CVD) in individuals with and without diabetes [8]. Large cohort studies have confirmed the strong and independent association between cardiovascular disease and CKD, including acute coronary syndrome (ACS), stroke, heart failure and sudden cardiac death, even after adjusting for known CVD risk factors, history of CVD events and proteinuria according to the stage of CKD based on eGFR [13]. Moreover, whilst individuals with a CKD stage of G5 (GFR < 15 mL/min/1.73 m^2^) or ESKF are at the highest individual risk of experiencing a cardiovascular event, due to the significantly higher prevalence of individuals in CKD stages G3a–G3b (eGFR 30–59 mL/min/1.73 m^2^), there will actually be more cardiovascular events occurring in this population [14,15,16].

## 2. Cystatin C

Cystatin C is a low-molecular-weight 122-amino acid and 12 kDA protein that is accumulating increasing interest. It is a member of the type 2 family of cysteine proteinase inhibitors. Cysteine proteinases irreversibly hydrolyse a peptide bond in an amino acid sequence and have a significant role in cell regulation, cell proliferation and adhesion, apoptosis, lipid metabolism and immune response [17].

The super family of cystatins are the predominant class of cysteine proteinase inhibitors. They are ubiquitously distributed in animals and enable regulation of the breakdown of both extracellular and intracellular proteins. As a result, they are key to ensuring balance between free cysteine proteinases and their complexes with inhibitors [18]. This allows for them to regulate against harmful cysteine proteinase activities, and as such, its concentrations have been linked to a number of other diseases. Cystatin C is encoded by a housekeeping C3T3 gene and is synthesised by all nucleated cells at a relatively constant rate [17]. It is broadly distributed and found in most body fluids, including plasma.

Interest in its role as a measure of kidney function stems from its physical and chemical properties as an endogenous marker. In addition to its constant rate of production, it is relatively freely filtered by glomeruli, is reabsorbed and catabolised by proximal renal tubular cells and is unaffected by muscle mass [19], in contrast to creatinine (Table 1).

## 3. Measuring Kidney Function

In 1976, Cockroft and Gault developed an equation which was correlated with creatinine clearance (CrCl), rather than measured GFR using exogenous markers [24]. This eGFR calculation eliminated the need for urine collection, but required multiple parameters (weight, age and gender), which were commonly not provided by the requesting healthcare workers. Further work led to new and more reliable eGFR equations; the Modification of Diet in Renal Disease (MDRD) study developed a 6-parameter equation (urine urea nitrogen excretion, serum urea, serum creatinine, age, gender and ethnicity). This MDRD6 equation was further refined with the substitution of the urine urea nitrogen excretion with serum albumin (MDRD7). Further modification led to a 4-parameter equation, removing the need for both albumin and serum urea concentrations to report an eGFR [25,26]. Importantly, this equation was aligned to measurement by isotope dilution mass spectrometry (IDMS), which is the gold-standard confirmatory test [27].

A major limiting factor of the MDRD equation is that it was developed in a cohort of people diagnosed with CKD; this led to a bias in calculation towards lower eGFR and increasing imprecision and underestimation of GFR at higher levels. Three years after the publication of the IDMS aligned 4-parameter MDRD equation, the Chronic Kidney Disease Epidemiology Collaboration published their equation, CKD-EPI_creat_ [28]. In 2020, the European Kidney Function Consortium (EKFC) published an equation with an extended age range from 2 to 90 years [29]. In 2021, further amendments were made to the CKD-EPI_creat_, providing an equation that did not require a modification factor based on race [30]. For most patients with CKD, the differences between these two equations are minor and are unlikely to lead to clinically significant differences.

The data from CKD-EPI demonstrated improved accuracy in GFR estimation when using both creatinine and cystatin C in combination (eGFR_creat-cys_) when compared to either marker alone. In the particular subgroup with eGFR_creat_ values 45–59 mL/min/1.73 m^2^, in which KDIGO also recommends cystatin C confirmatory testing, the combined equation correctly reclassified 16.8% of those with eGFR 45–59 mL/min/1.73 m^2^ to eGFR > 60 mL/min/1.73 m^2^ [31]. The removal of the diagnosis and label of CKD may be reassuring to patients and could potentially help clinicians to focus their efforts on patients with higher-risk CKD.

## 4. Bias, Accuracy and Correlation

There are clear advantages of using Cystatin C as a measure of renal function. However, how does this impact on the bias, accuracy and correlation of eGFR equations that incorporate this marker either in isolation (eGFR_cys_) or in combination with creatinine (eGFR_cys/creat_)? That is, how do these eGFR values compare to those measured GFR values (inulin, iohexol and iothalamate)?

A comparison of the statistical P30 testing (the % of samples with eGFR within 30% of the confirmed measured GFR) showed that over time, accuracy was improving with subsequent iterations of the eGFR equation. The accuracy of the Cockroft Gault, MDRD and CKD-EPI, using creatinine, were 74.2, 81.2 and 84.5%, respectively [32]. A comparison of bias showed that CKD-EPI_cys_ was 4.84 (mL/min/1.73 m^2^) lower than CKD-EPI_creat_ and 1.5 lower than using CKD-EPI_creat/cys_. These gaps increased in subgroups of low measured GFR (<60 mL/min/1.73 m^2^). CKD-EPI_creat/cys_ achieved the highest accuracy—7.5% higher than CKD-EPI_creat_ and 3.21% higher than CKD-EPI_cys_. This metanalysis also concluded that both the Pearson correlation coefficient and Fisher’s z-transformed R were the highest between measured GFR and CKD-EPI_creat/cys_ (Table 2) [33].

## 5. Analytical Performance

In practice, serum creatinine is widely used in the majority of patients who have blood tests measured routinely, and therefore, many clinicians will likely feel comfortable interpreting the results. Concentrations are now routinely measured using enzymatic methods, although the older colorimetric Jaffe reaction is still in use. A reference material (SRM 967, National Institute for Standards and Technology, Washington, DC, USA) is available for creatinine and most widespread methods are calibrated against the IDMS to reduce error and maximise comparability of measurements. Due to the more recent introduction of a reference material (ERM-DA471/IFCC—2011) for cystatin C, the measurements of all methods are now traceable; manufacturers have markedly improved accuracy and the between-method agreements of cystatin C measurements since 2014, thus allowing for greater confidence in estimating GFR, which relies on cystatin C [34].

The mainstay of cystatin C measurement is through particle-enhanced immunoassays, via either a turbidometric immunoassay (PETIA) or nephelometric immunoassay (PENIA) (Table 3). The advantage of using these assays is a fast turnaround time, lack of interferences by other substances and higher precision. Both assays were reported to be less likely influenced by substances that affect creatinine measurement (haemolysis, lipaemia and icterus) [35]. It has been reported that the PENIA method is more sensitive than the PETIA method and should be considered as the method of choice when measuring cystatin C [36]. Furthermore, unlike creatinine, cystatin C is unaffected by race or genetic ancestry, which is one of the reasons for the recent recommendation from the National Kidney Foundation and American Society of Nephrology to increase the use of cystatin C to estimate kidney function [37]. However, cystatin C levels may be correlated with oxidative stress and inflammation and may be slightly higher in patients with cardio-metabolic conditions, thyroid disease, malignancy or glucocorticoid therapy [33].

## 6. Clinical Applications

Studies have demonstrated that the cystatin-C-based estimates of GFR are more effective predictors of clinical outcomes than creatinine-based eGFR and identify more patients who are at increased risk in elderly [39,40,41,42,43] and non-white ethnic groups [44]. This is particularly evident in cases of mortality and cardiovascular disease, where the superiority of cystatin C is most pronounced among individuals with GFRs greater than 45 mL/min/1.73 m^2^ [5,45,46,47,48,49]. Furthermore, recent research has shown that incorporating cystatin C with creatinine leads to more accurate GFR estimation and improves the classification of CKD [37,47].

### 6.1. Elderly

The Health, Aging, and Body Composition cohort study, focusing on elderly patients aged 70–79 years, revealed a significant association between increasing levels of cystatin C and an elevated risk of mortality. This association remained consistent for both cardiovascular mortality and mortality from other causes, but not for cancer-related mortality. Conversely, after adjusting for a range of factors, creatinine levels were not found to be associated with mortality. Therefore, cystatin C emerged as a robust and independent risk factor for mortality in the elderly population [39].

In another study called the Cardiovascular Health Study, which involved over 4000 patients who were elderly without chronic kidney disease (CKD), eGFR (estimated glomerular filtration rate) was measured using the MDRD equation and cystatin C. The study found strong associations between cystatin C concentrations and several adverse outcomes, including death (hazard ratio (HR 1.33), incident heart failure (HR 1.28), stroke (HR 1.22), and myocardial infarction (HR 1.20). On the other hand, serum creatinine concentrations exhibited weaker associations with each of these outcomes and only predicted cardiovascular death. Participants without CKD but with elevated cystatin C concentrations had a four-fold risk of progressing to CKD within a four-year follow-up period [40,41,42,43].

### 6.2. Ethnicity

As previously mentioned, unlike creatinine, cystatin C is unaffected by race or genetic ancestry. Supporting this, a large prospective study on individuals who are South Asian in the UK Biobank by Debbie Chen [44] found that cystatin C identified more than five times the number of participants with eGFR < 60 mL/min 1.73 m^2^ when compared with creatinine, and eGFR_cys_ was on average lower than eGFR_creat_ by 13 mL/min 1.73 m^2^. This study identified a CKD population at elevated risk for mortality, incident heart failure, and incident atherosclerotic cardiovascular disease that was not detected by creatinine.

More research is needed to identify other high-risk ethnic groups where cystatin C measurements may help identify and stratify the risk of mortality and morbidity.

### 6.3. Cardiovascular Disease

Multiple studies have found that eGFR_cys_ and eGFR_cys-creat_ below 85 mL/min/1.73 m^2^ were associated with increased risks of death from any cause and cardiovascular disease. Current guidelines and practice in CKD define a threshold of 60 mL/min/1.73 m^2^ with eGFR_creat_, suggesting that cystatin C or a combination calculation eGFR can detect patients at risk of cardiovascular disease, particularly patients with deteriorating renal function much earlier, and well above what is currently considered normal renal function [6,7,50].

In a comprehensive study conducted within the UK Biobank, which included over 400,000 participants, the addition of both eGFR_cys_ and eGFR_creat-cys_ to traditional risk factors for atherosclerotic diseases significantly enhanced the accuracy of predictions for all-cause mortality, as well as fatal and non-fatal cardiovascular disease [5,46]. Notably, the greatest improvement in discrimination was observed with eGFR_cys_, whereas the inclusion of eGFR based on creatinine did not contribute to improved discrimination. In this particular cohort study, patients with an estimated glomerular filtration rate based on serum creatinine levels below 60 mL/min/1.73 m^2^ exhibited either low or high risks of future cardiovascular disease and mortality. Conversely, eGFR based on cystatin C effectively stratified the risk levels in these individuals. These findings indicate that, without cystatin C testing, eGFR alone inadequately distinguishes the broader risks associated with mild chronic kidney disease based solely on serum creatinine levels. Irrespective of the eGFR based on creatinine, the 10-year probabilities of cardiovascular disease and mortality were low when the eGFR_cys_ was equal to or greater than 60 mL/min/1.73 m^2^. However, with eGFR_cys_ below 60 mL/min/1.73 m^2^, the 10-year risks were nearly doubled in older adults and more than doubled in younger adults. The use of eGFR_cys_ demonstrated better discriminatory ability for assessing the risk of cardiovascular disease and mortality compared to eGFR based on creatinine. It is important to note that these findings pertain specifically to individuals with a normal urine albumin concentration (<30 mg/mmol), as outcomes were not compared in participants with albuminuria exceeding 30 mg/mmol.

In a separate analysis conducted on participants with CKD compared to those without CKD in the MESA (multi-ethnic study of atherosclerosis) and CHS (cardiovascular health study), adjusted hazard ratios for mortality were assessed. In the MESA study, the hazard ratio for mortality was 0.80 for CKD identified by creatinine only, 3.23 for CKD identified by cystatin C only, and 1.93 for CKD identified by both markers. Similarly, in the CHS study, the adjusted hazard ratios were 1.09, 1.78, and 1.74, respectively. This pattern was consistent across various outcomes, including cardiovascular disease, heart failure, and kidney failure. It was observed that an unfavourable prognosis was specifically associated with the subset of individuals who were identified as having CKD according to the cystatin C-based equation [48].

### 6.4. Reclassification

Dharnidharka conducted a meta-analysis of 54 studies to assess the accuracy of cystatin C and creatinine in comparison to the reference standard of GFR [45]. The results showed that the overall correlation coefficient for the reciprocal of serum cystatin C (r = 0.816) was higher than that of serum creatinine (r = 0.742; *p* < 0.001). Additionally, when evaluating the ROC-plot area-under-the-curve values, cystatin C demonstrated a closer resemblance to the reference test for GFR (0.926) compared to creatinine (0.837; *p* < 0.001). Notably, immunonephelometric methods of the cystatin C assay exhibited significantly stronger correlations than other assay methods (r = 0.846 versus r = 0.784; *p* < 0.001).

Cystatin C demonstrated the ability to reclassify patients in CKD 3A categories to milder or non-CKD levels of eGFR, which can have several positive impacts. This reclassification may alleviate the burden on healthcare systems, reduce unnecessary referrals in certain patient groups and allow for healthcare efforts to be focused on individuals at the highest risk. Peralta, in a study conducted in 2011, proposed the development of a triple-marker panel utilizing cystatin C in combination with creatinine and albuminuria [47]. In their prospective analysis of 26,000 US adults enrolled in the Reasons for Geographic and Racial Differences in Stroke (REGARDS) study, the use of this panel enabled the stratification of patients into higher and lower risk groups. The subset of participants who had CKD defined by all three markers showed a concentrated risk of future end-stage renal disease. Additionally, there was a second-highest risk group for end-stage renal disease that was missed by creatinine alone but detected by cystatin C and the albumin–creatinine ratio.

In cases where patients had a normal eGFR_cys_ but a decreased eGFR_creat_, their risk level was similar to individuals with a normal eGFR. The author suggests that specifically targeting patients with CKD as defined by creatinine and assessing their cardiovascular risk would require only a small number of cystatin C tests. However, on the other hand, participants who were classified as having a decreased eGFR_cys_ but not eGFR_creat_ were also found to be at a higher risk for adverse events. Detecting these individuals would necessitate screening a significantly larger number of people compared to confirming a decreased GFR alone.

## 7. Cost and Implementation

There are three main obstacles to incorporating routine cystatin C testing: the cost, accessibility, and the level of clinical awareness and understanding of test results. Cystatin C testing is approximately ten times more costly than creatinine testing, with a price of GBP 2.50 (USD 3.00) per test compared to GBP 0.25 (USD 0.30) for creatinine [50]. Despite this significant additional expense, cystatin C testing is comparable or even less expensive than other commonly conducted tests for patients with CKD, such as parathyroid hormone, C-reactive protein, and vitamin D [51].

In a retrospective analysis of primary care laboratory requests in Oxfordshire, England, which represented a population of 600,000 individuals, 22,240 people with stable stage 3a CKD and no proteinuria were identified [7]. Since the population of Oxfordshire comprises approximately 1% of the total UK population, it implies that there is an initial need to determine the CKD status of at least 2 million individuals using cystatin C testing. Among the UK laboratories, only eight (2.1% of total) reported providing cystatin C assays. This highlights a significant disparity between the demand for cystatin C testing in primary care and the availability of national assay services. Consequently, this discrepancy poses a major obstacle to the implementation of NICE guidance.

## 8. Conclusions

Cystatin C has advantages over creatinine as a marker of kidney function. It is produced at a constant rate by all nucleated cells, and freely filtered by glomeruli, making it less susceptible to individual patient characteristics such as age, gender and muscle mass. Moreover, cystatin C is not affected by non-kidney factors such as diet, drugs and inflammation, which can affect creatinine levels. As a result, cystatin C is a more reliable and sensitive marker of renal function, especially in populations where creatinine measurements may be less accurate.

A growing body of evidence supports the use of cystatin C in the diagnosis and management of chronic kidney disease. Studies have shown that cystatin C is more accurate than creatinine in detecting the initial stages of chronic kidney disease and in predicting the risk of adverse outcomes such as kidney failure and death. Furthermore, cystatin C has been shown to be a better predictor of cardiovascular disease and mortality than creatinine, suggesting that it may have broader clinical utility beyond renal function assessment.

Given the advantages of cystatin C over creatinine, national efforts should be made to improve the availability and accessibility of routine cystatin C testing. Unpublished data (United Kingdom National External Quality Assessment Service—UK NEQAS) shows that external quality assessment samples are provided for creatinine and cystatin C to 605 and 17 laboratories, respectively. This is despite all major platform manufacturers providing access to their own cystatin C assay or allowing for third party assays to be run on their analysers.

This will be especially important for patients with mild kidney disease of eGFR 45–59 mL/min/1.73 m^2^ (G3a), who may be missed by creatinine-based estimates of GFR, and for patients who are at increased risk of kidney disease due to factors such as diabetes, hypertension, family history or deteriorating renal function. Additionally, patients of non-white or mixed ethnic backgrounds, as well as those with increasing age or frailty, may benefit from cystatin C testing, as these factors can affect creatinine levels and lead to inaccurate estimates of GFR. Finally, patients at weight or raw muscle mass extremes, including those with dystrophic diseases, may also benefit from cystatin C testing, as these conditions can affect creatinine production and clearance.

## Figures and Tables

**Table 1 biomolecules-13-01075-t001:** Summary of cystatin C and creatinine function, processing, filtration, and influencing factors and advantages [17,18,20,21,22,23].

	Creatinine	Cystatin C
Function	Heterocyclic nitrogenous compound 113 kDA. Produced from creatine in muscle. No physiological function known.	Endogenous 13 kDa protein. Synthesised by all nucleated cells. Cysteine protease inhibitor.
Renal Processing	Excreted unchanged by kidneys, majority via glomerular filtration but a small amount by active secretion.	Relatively freely filtered by glomeruli. Reabsorbed and catabolised by proximal renal tubular cells.
Basal Production	Constant.	Constant.
Decreasing Factors	Increasing age, female gender, Asian ethnicity. Inflammation. Neuromuscular illness or amputation.	Hypothyroidism.
Increasing Factors	Black ethnicity. Increased muscle mass. High protein diet and supplements.	Steroid treatment. Current cigarette smoking. Chronic inflammation. Obesity. Hyperthyroidism.
Advantages	More routinely available. Standardised assays. Health care professionals more comfortable with its use.	Not dependent on muscle mass, physical activity or protein intake. Less influenced by age or gender. Rises faster than creatinine (earlier detection).

**Table 2 biomolecules-13-01075-t002:** Comparison of the correlation of CKD-EPI_creat_, CKD-EPI_cys_ and CKD-EPI_creat/cys_ eGFR calculated values with measured GFR values. CI—Confidence Interval.

CKD-EPI eGFR (With)	Pearson Correlation	95% CI	Fisher’s z-Transformed Correlation	95% CI
Creatinine	0.77	0.69–0.86	1.07	0.79–1.35
Cystatin	0.76	0.68–0.85	1.04	0.77–1.32
Creatinine and Cystatin	0.81	0.73–0.89	1.2	0.89–1.50

**Table 3 biomolecules-13-01075-t003:** Comparison of analytical methods and performance of creatinine and cystatin C [38]; CV—coefficient of variance, RCV—reference change value, SRM—standard reference material, M—male, F—female, PETIA—particle-enhanced turbidimetric immunoassay, PENIA—particle-enhanced nephelometric immunoassay, IDMS—isotope-dilution mass spectrometry.

	Creatinine	Cystatin C
Analytical Method	Chemical Enzymatic IDMS	PETIA PENIA
Reference Method	None	No single reference method stated
Reference Material	SRM 967	ERM-DA471/IFCC
Sources of Error	To reduce error, most methods are now calibrated against IDMS	Earlier methods (before 2011) not traceable
Interfering Substances	Spectral and chemical interferences vary between manufacturers and methods Haemolysed samples, lipaemia, icteric samples	
Reference Interval (Adults)	60–120 μmol/L (M) 55–100 μmol/L (F)	Age and method specific, common reference range between 1 and 50 years Typical interval of 0.6–1.1 mg/L
Interindividual CV	10%	49.9%
Intraindividual CV	4.6%	4.8%
Index of Individuality	0.46	0.1
Analytical CV	6.3%	0.9%
RCV Positive	22%	14%
RCV Negative	22%	13%

## Data Availability

Data sharing not applicable; no new data were created or analyzed in this study. Data sharing is not applicable to this article.

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
