# Peer review of "Should Cystatin C eGFR Become Routine Clinical Practice?"

_biomolecules, 2023, doi:10.3390/biom13071075_

Round 1
Reviewer 1 Report
The review by Spencer and colleagues covers the important and current discussion on cystitis C use for eGFR estimation. The paper is overall well written and organised however I do believe Authors should discuss in a separate paragraph the studies investigating the associations between cystitine-C and GFR calculated by the gold standard inulin clearance. There are few studies analysing these associations and I believe they should be included to provide demonstration of cystitis-C-eGFR equation accuracy.
Author Response
Many thanks for your feedback.
Although many of the initial studies did indeed use inulin as an endogenous marker (“Gold Standard”), more recent good quality studies of the correlation between eGFR and mGFR have used both iohexol and iothalamate, which have supplanted inulin.
We have expanded the paragraph (line 134 – 142) which explores the bias and accuracy of eGFR equations to include the results of a robust, good quality metaanalysis. This incorporates all measures of mGFR and calculated Peason’s correlation coefficient CKD-EPI vs mGFR.
Reviewer 2 Report
With this review, Spencer et al. have addressed an extremely important topic in nephrology as well as cardiology. They have structured the review logically and prepared it conscientiously. The clear tables illustrate the advantages of the Cystatin C eGFR. I have no comments or suggestions for improvement and fully recommend the review for acceptance for publication.
Author Response
Many thanks for your feedback.
Reviewer 3 Report
It is a well organized review article, I don't have any concerns
Author Response
Many thanks for your feedback.
Reviewer 4 Report
The authors review the advantage of cystatin C eGFR for routine clinical practice instead creatinine eGFR. Although the cost of cystatin C method for evaluation kidney function compared to creatinine, earlier finding of cardiovascular risk of patients overcome this advantage. Also, there are some groups of patients that creatinine eGFR should not be applicable as some race and older. So, I recommend this article to be published.
Author Response
Many thanks for your feedback.
Round 2
Reviewer 1 Report
Authors provided evidence for mGFR vs. eGFR and the paper is now suitable for publication